# ReGenText: Joint Generation and Restoration for Diverse Text Image Super-Resolution

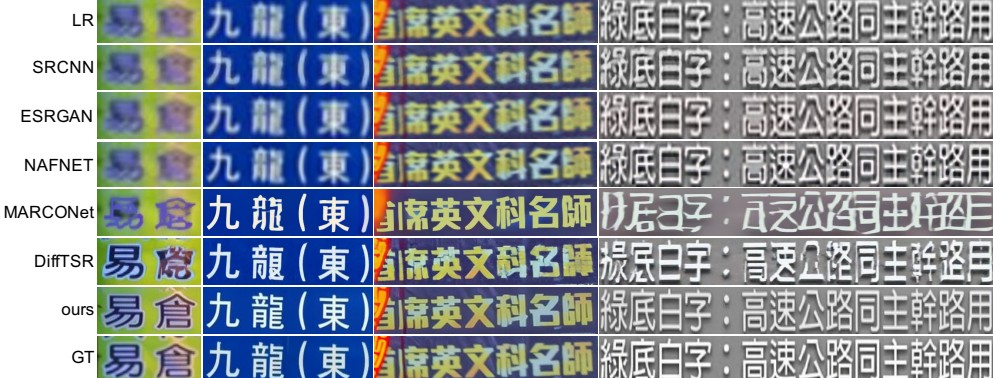

Figure 1: Text image super-resolution results of our method on diverse scene texts.

## Abstract

Text Image Super-Resolution (TISR) aims to recover high-resolution images from low-resolution inputs degraded by unknown factors. The goal is to produce visually faithful output while preserving text readability and semantic consistency. Despite recent progress, existing methods struggle to preserve structure and visual fidelity under complex glyphs, severe degradations, or varied layouts. This mainly stems from three challenges: lack of training data, limitations in model fidelity, and limited adaptability to complex layouts. Therefore, we novelly propose **ReGenText**, a systematic solution for diverse text super-resolution. ReGenText integrates data generation, image restoration, and training strategies, substantially mitigating the three aforementioned challenges. Specifically, we propose: **Gen-Text**: a diffusion-based data generation framework that combines font styles and glyph priors to synthesize large-scale, high-quality text images, effectively alleviating data scarcity; **Re-Text**: a hybrid diffusion–GAN model that balances structural precision and fine-detail restoration for high-fidelity reconstruction; **Bucket Training**: a training strategy that groups samples based on text length and orientation, improving generalization on long and vertical texts. Extensive experiments show that ReGenText achieves state-of-the-art performance in both text recognition and visual quality across multiple benchmarks.

## 1 Introduction

Text images serve as crucial information carriers in computer vision (Chen et al., 2021b; Ma et al., 2023c), where their clarity and readability directly affect the performance of downstream tasks (Zhang et al., 2024; Jiang et al., 2024; Chen et al., 2025; Hu et al., 2024). Unlike natural images, text images demand exceptionally high visual fidelity. Even minor stroke omissions or structural distortions may result in significant semantic errors (Especially in Chinese). Therefore, Text Image Super-Resolution (TISR) not only restores high-quality visual appearance but also faithfully preserves character structures and semantics.

However, existing methods (Ma et al., 2023b; 2022; Li et al., 2023) still struggle to restore text structure and maintain visual fidelity in real-world scenarios, as shown in Fig. 1. Traditional text super-resolution methods (Ma et al., 2023b; 2022) have limited generative capacity and struggle to

recover text structures from severely degraded images, even with structural priors or recognition modules. Although recent methods (Li et al., 2023; Zhang et al., 2024) leverage GANs and diffusion models to improve performance, these methods are prone to introducing excessive texture hallucinations and instability under diverse layouts (e.g., horizontal or vertical).

The reasons can be attributed to the following three aspects: **Lack of training data.** The scale of real-world text image datasets (Chen et al., 2021b) remains limited, with fewer than 100,000 high-quality samples available. Moreover, complex fonts and rare characters are even more scarce, making it difficult for models to learn diverse glyph structures. **Limitations in model fidelity.** Insufficient generative capacity fails to restore text structure; conversely, excessive capability introduces textural hallucinations (Li et al., 2022), compromising visual fidelity. **Limited adaptability to complex layouts.** Current models (Zhang et al., 2024), constrained by fixed-size input designs during training, struggle to handle complex layout scenarios such as long texts and vertical texts. To tackle these challenges, there is an urgent need for an approach that integrates data, models, and training strategies to ensure high-fidelity restoration of text images.

Therefore, we propose **ReGenText**, a systematic solution for diverse text image super-resolution, as shown in Fig. 2. Unlike previous works that focus on model refinements, ReGenText systematically integrates data generation, image restoration, and training strategies to address the three key bottlenecks. Specifically, ReGenText builds on three complementary perspectives: **1). Data generation with Gen-Text.** We develop Gen-Text, a generation network that integrates reference style guidance with glyph priors. Gen-Text significantly improves style consistency and stroke-level structure accuracy, enabling the synthesis of large-scale, high-quality text image samples. **2). Image restoration with Re-Text.** We propose Re-Text, a novel hybrid architecture that combines pixel-level diffusion (Hoogeboom et al., 2023) with GANs. Re-Text integrates the text generation capabilities of diffusion models with the detail-sharpening strengths of GANs, achieving a complementary improvement in both visual fidelity and text clarity. **3). Layout-aware learning with Bucket Training.** To handle the diversity of text layouts, ReGenText introduces a bucket training strategy that groups samples by length and layout, improving robustness on long and vertical texts.

Subsequent experimental results show that on the CTR-TSR-Test benchmark (Chen et al., 2021b), our method improves OCR accuracy by 7.08%. In addition, we extend existing scene text benchmarks by incorporating variable-length and vertical texts, and further construct a rare-character dataset to systematically evaluate model generalization. The results show that ReGenText improves accuracy by 5.19% in variable-length texts and 14.05% in rare character texts, significantly outperforming existing methods. The main contributions of our work are summarized as follows:

- We propose ReGenText, a systematical paradigm that integrates data generation, model design, and training strategies, effectively allevi- ating three key challenges in text image super-resolution: limited training data, compromised visual fidelity, and poor adaptability to complex layouts.

- ReGenText builds on three pillars: Gen-Text: integrates font style and glyph priors to synthesize large-scale, high-fidelity training samples, fundamentally alleviating data scarcity; Re-Text: introduces a hybrid diffusion–GAN architecture for dual high-fidelity restoration of structure and details; Bucket Training: employs a length- and layout-aware adaptive grouping strategy to enhance generalization under complex layouts.

- We extend existing scene text benchmarks to include variable-length and vertical text, forming CTR-X, and further construct a dataset of rare Chinese characters, named RareText. This provides a testing platform for evaluating its ability to generate complex glyphs.

- In multiple benchmarks, our proposed model consistently outperforms existing methods in both text recognition accuracy and perceptual quality metrics, achieving state-of-the-art performance. These results highlight the effectiveness and robustness of ReGenText.

## 2 RELATED WORK

### 2.1 TEXT IMAGE GENERATION

Text image synthesis plays a key role in scene text recognition, enhancement, and generation. Early methods (Jaderberg et al., 2014; Gupta et al., 2016) rely on text rendering engines or graphics

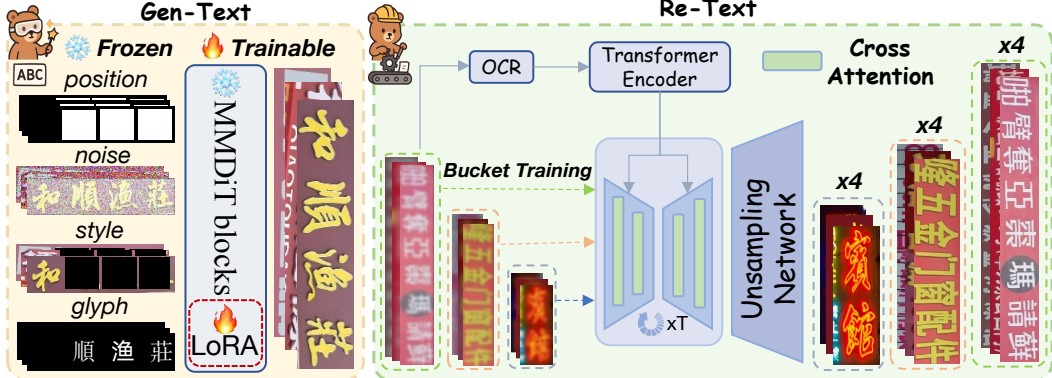

Figure 2: We present **ReGenText**, a systematic solution for text super-resolution via data generation, image restoration, and bucket training. Gen-Text can synthesize large-scale, high-fidelity training samples; Re-Text achieves dual high-fidelity of text structure and visual details.

pipelines to generate characters from font libraries and overlay them onto backgrounds. However, these methods exhibit significant gaps in texture, shadow, and background integration compared to real-world scenes. To overcome these limitations, recent work has increasingly adopted generative models for text image synthesis. UDiffText (Zhao & Lian, 2024) and Glyph-ByT5 (Liu et al., 2024a;b) improve glyph awareness by aligning text encoders with visual features. GlyphDraw (Ma et al., 2023a), GlyphControl (Yang et al., 2023), and AnyText (Tuo et al., 2023) leverage glyph structures to guide generation. Nevertheless, architectural limitations and complex backgrounds still challenge these methods in preserving font style consistency and stroke accuracy. To address this, we propose Gen-Text, a Diffusion Transformer (Peebles & Xie, 2023) for text image generation. Gen-Text integrates style information and glyph representations as dual constraints in the diffusion process, improving both font consistency and stroke fidelity. By generating highly realistic text lines, it provides super-resolution models with abundant high-quality training data.

## 2.2 TEXT IMAGE SUPER-RESOLUTION

Although natural image super-resolution (Wang et al., 2024; Wu et al., 2024a;b) has achieved remarkable progress, it often falls short when applied to text images. This is mainly because text images demand high structural integrity and stroke-level precision. To address this gap, Text Image Super-Resolution (TISR) (Wang et al., 2019; Mou et al., 2020; Wang et al., 2020; Zhao et al., 2022) has been proposed. TPGSR (Ma et al., 2023b) incorporates text priors to enhance glyph representations, while TATT (Ma et al., 2022) introduces a global attention module to better handle irregular text layouts. C3-STISR (Zhao et al., 2022) leverages triple cues: recognition feedback, visual cues, and language cues, to guide text restoration. MARCONet (Li et al., 2023) employs a generative structure prior by combining a structural codebook with style latents under the StyleGAN framework, enabling the recovery of diverse character styles. DiffTSR (Zhang et al., 2024) integrates image and text diffusion models and utilizes a multi-modal fusion module to simultaneously refine style and structure during the diffusion process. However, these methods still struggle to achieve simultaneous structural and visual fidelity and perform poorly on variable-length text. To address this, we propose Re-Text, a hybrid architecture that combines the text generation capability of diffusion models with the detail refinement strength of GANs. Moreover, we introduce a bucket training strategy, which significantly enhances the model's generalization ability on complex text layouts.

## 3 METHOD

To tackle the challenges of data scarcity, structural fidelity, and layout diversity in text image super-resolution, we propose ReGenText, a systematic framework comprising four key components. 3.1 **Gen-Text**: generates diverse, high-quality text images using font styles and glyph priors to mitigate data scarcity. 3.2 **Re-Text**: restores fine-grained structures and visual fidelity in degraded text images via a hybrid diffusion–GAN model. 3.3 **Bucket Training**: groups samples by text length and layout to improve robustness on long and vertical texts. 3.4 **Benchmark**: provides CTR-X and RareText, evaluating models on variable-length, vertical, complex, and rare characters.

### 3.1 GEN-TEXT FOR TEXT IMAGE GENERATION

A key limitation of existing text image super-resolution models is the scarcity of training data, particularly for rare and complex characters. To address this, we propose Gen-Text, a diffusion-based generation network that leverages font styles and glyph priors to synthesize high-fidelity training samples with both semantic and visual consistency.

Specifically, we denote the input image as $x \in \mathbb{R}^{H \times W \times 3}$, the position binary mask as $m \in \{0, 1\}^{H \times W}$, the glyph image as $g \in \mathbb{R}^{H \times W \times 3}$, and the textual prompt as $y$. We encode $x$, $g$, and $m$ with a VAE (Kingma & Welling, 2013) to obtain the latent features $z_0$, $z_g$, and $z_m$. The style feature is obtained by masking out the foreground, $x \odot (1 - m)$, and encoding the result with the VAE to produce $z_s$. Following the rectified flow (RF) paradigm (Esser et al., 2024), Gaussian noise $\epsilon_g \sim \mathcal{N}(0, I)$ is sampled, and for each latent a random timestep $t$ is drawn according to a density $p(t)$. A flow-matching noise scale $\sigma(t)$ is computed, yielding the noisy latent:

$$z_t = (1 - \sigma(t)) z_0 + \sigma(t) \epsilon_g, \quad where \ t \sim p(t), \epsilon_g \sim \mathcal{N}(0, I). \tag{1}$$

The latent feature $z_t$ is concatenated with $z_m$ and $z_s$ along the channel dimension, denoted as $z_k = Concat(z_t, z_m, z_s)$. Meanwhile, the T5 encoder (Raffel et al., 2020) is employed to encode the text condition $y$, producing the text embedding $c_{te}$. And then, a DiT denoiser $\epsilon_\theta(\cdot)$ is employed to predict the noise added to the latent image $z_t$ with the following objective:

$$\mathcal{L}_d = \mathbb{E}_{z_0, z_k, z_g, c_{te}, t} \left[ \| \epsilon_g - \epsilon_\theta(z_k, z_g, c_{te}, t) \|_2^2 \right]. \tag{2}$$

Here, $\hat{x}$ denotes the predicted image obtained by removing the noise from the latent $z_t$ using the DiT denoiser $\epsilon_\theta(\cdot)$ and subsequently decoding it through the VAE. To explicitly preserve glyph structures, we introduce a Sobel loss (Roberts & Mullis, 1987) that encourages the gradients of the generated image to align with those of the ground truth, effectively maintaining stroke edges:

$$\mathcal{L}_{sobel} = \|\text{Sobel}(\hat{x}) - \text{Sobel}(x)\|_1, \tag{3}$$

The final training objective jointly optimizes both the RF loss and the Sobel loss:

$$\mathcal{L}_{gen} = \mathcal{L}_d + \lambda_{\text{Sobel}} \mathcal{L}_{\text{Sobel}}. \tag{4}$$

Through these operations, Gen-Text generates text images with precise structures and diverse styles, effectively expanding the training data and mitigating the long-tail problem for rare characters.

### 3.2 RE-TEXT FOR TEXT IMAGE SUPER-RESOLUTION

Although Gen-Text alleviates the problem of limited training data, restoring visual fidelity and textual structure remains challenging when handling severely degraded text images. Therefore, we propose Re-Text, which performs diffusion directly in the pixel space, and further combine it with a GAN-based refinement to enhance perceptual quality.

Specifically, given a high-resolution text image $x_{HR}$, we first apply Real-ESRGAN (Wang et al., 2021) to generate the corresponding low-resolution image $x_{LR}$. To improve modeling efficiency, both $x_{LR}$ and $x_{HR}$ are further downsampled by a factor of 4 using bilinear interpolation, and the results are fed into the diffusion model. Following the DDPM Ho et al. (2020) paradigm, Gaussian noise is gradually added to $x_{HR}$:

$$x_t = \sqrt{\alpha_t} \, x_{HR} + \sqrt{1 - \alpha_t} \, \epsilon_r, \quad \epsilon_r \sim \mathcal{N}(0, I), \tag{5}$$

where $\alpha_t$ denotes the variance schedule at time step $t$. The resulting noisy latent $x_t$ is concatenated with the original $x_{LR}$ along the channel dimension. In addition, the text prior $c$, obtained from OCR predictions, is incorporated via a cross-attention module, providing semantic guidance to the generation process. The diffusion loss is defined as:

$$\mathcal{L}_{diff} = \mathbb{E}_{x_0, x_{LR}, \epsilon_r, c, t} \left[ \| \epsilon_r - \epsilon_\theta(x_t, x_{LR}, c, t) \|_2^2 \right], \tag{6}$$

where $\epsilon_\theta$ is the denoiser parameterized by the diffusion model.

To further improve sharpness and perceptual realism, we introduce an adversarial refinement stage with a generator $G$ and a discriminator $D$. $G$ reconstructs high-resolution images from noise-corrected features conditioned on the low-resolution input, while $D$ encourages $G$ to produce outputs indistinguishable from real high-resolution images.

Table 1: Quantitative comparison with state-of-the-art methods on CTR-TSR-Test/ReaCE x4.

| Method | CTR-TSR-Test | | | | | RealCE | | | | |
|---|---|---|---|---|---|---|---|---|---|---|
| | PSNR ↑ | LPIPS ↓ | FID ↓ | ACC ↑ | NED ↑ | PSNR ↑ | LPIPS ↓ | FID ↓ | ACC ↑ | NED ↑ |
| SRCNN | 20.74 | 0.501 | 116.5 | 0.6031 | 0.6160 | 16.63 | 0.364 | 128.1 | 0.7101 | 0.8018 |
| ESRGAN | 20.90 | 0.310 | 21.86 | 0.6179 | 0.6272 | 16.84 | 0.407 | 83.22 | 0.7121 | 0.8047 |
| NAFNET | 21.82 | 0.447 | 87.93 | 0.6451 | 0.6573 | 16.76 | 0.359 | 118.1 | 0.7122 | 0.8023 |
| TSRN | 19.41 | 0.535 | 137.3 | 0.6149 | 0.6267 | 15.22 | 0.418 | 148.5 | 0.6963 | 0.7873 |
| TBSRN | 21.56 | 0.442 | 132.6 | 0.6360 | 0.6486 | 16.51 | 0.367 | 130.8 | 0.7050 | 0.7960 |
| TATT | 21.84 | 0.453 | 107.6 | 0.6273 | 0.6403 | 16.79 | 0.422 | 118.3 | 0.7214 | 0.8135 |
| MARCONet | 19.33 | 0.436 | 108.5 | 0.5123 | 0.5241 | 16.04 | 0.397 | 103.1 | 0.6638 | 0.7411 |
| DiffTSR | 21.85 | 0.231 | 8.482 | 0.8350 | 0.8471 | 17.49 | 0.336 | 70.59 | 0.8475 | 0.8747 |
| **Ours** | **26.73** | **0.163** | **8.067** | **0.9058** | **0.9075** | **19.61** | **0.301** | **55.49** | **0.8791** | **0.8801** |

Table 2: Quantitative comparison with state-of-the-art methods on CTR-X/RareText x4.

| Method | CTR-X | | | | | RareText | | | | |
|---|---|---|---|---|---|---|---|---|---|---|
| | PSNR ↑ | LPIPS ↓ | FID ↓ | ACC ↑ | NED ↑ | PSNR ↑ | LPIPS ↓ | FID ↓ | ACC ↑ | NED ↑ |
| SRCNN | 25.03 | 0.370 | 83.03 | 0.5658 | 0.5760 | 23.21 | 0.414 | 111.2 | 0.4500 | 0.4607 |
| ESRGAN | 24.74 | 0.450 | 87.51 | 0.5658 | 0.5679 | 22.85 | 0.466 | 108.7 | 0.4530 | 0.4631 |
| NAFNET | 22.71 | 0.4084 | 81.43 | 0.6034 | 0.6048 | 21.83 | 0.422 | 103.5 | 0.4600 | 0.4696 |
| TSRN | 19.12 | 0.4697 | 88.17 | 0.5658 | 0.5679 | 18.58 | 0.452 | 103.6 | 0.4214 | 0.4300 |
| MARCONet | 11.04 | 0.771 | 105.8 | 0.5319 | 0.5349 | 13.52 | 0.659 | 129.4 | 0.3635 | 0.3723 |
| DiffTSR | 20.97 | 0.329 | 50.72 | 0.5949 | 0.5961 | 19.58 | 0.327 | 65.62 | 0.4054 | 0.4145 |
| **Ours** | **26.97** | **0.165** | **29.05** | **0.6468** | **0.6488** | **24.48** | **0.221** | **46.25** | **0.5459** | **0.5560** |

Specifically, the predicted noise $\epsilon_\theta$ from the diffusion denoiser is first used to remove the noise from the noisy latent $x_t$:

$$\hat{x}_{HR} = x_t - \sqrt{1-\alpha_t}\,\epsilon_\theta(x_t, x_{LR}, c, t), \tag{7}$$

where $\hat{x}_{HR}$ denotes the noise corrected feature. The generator then reconstructs the final high-resolution image. The adversarial objectives for the discriminator and generator are defined as:

$$\mathcal{L}_{gan}^D = -\mathbb{E}_{x_{HR}}\big[\log D(x_{HR})\big] - \mathbb{E}_{\hat{x}_{HR}}\big[\log(1 - D(\hat{x}_{HR}))\big], \quad \mathcal{L}_{gan}^G = -\mathbb{E}_{\hat{x}_{HR}}\big[\log D(\hat{x}_{HR})\big], \tag{8}$$

where $\mathcal{L}_{gan}^D$ updates the discriminator, and $\mathcal{L}_{gan}^G$ provides the adversarial refinement signal for the generator. The overall training loss can be expressed as:

$$\mathcal{L}_{re} = \mathcal{L}_{diff} + \lambda_{\text{gan}} \mathcal{L}_{gan}^G. \tag{9}$$

This hybrid framework enables high-quality text image super-resolution with both structural preservation and realistic visual appearance.

### 3.3 BUCKET TRAINING STRATEGY

To improve generalization across varying text lengths and layouts, we propose the Bucket Training strategy to both Gen-Text and Re-Text. For super-resolution models, previous methods (Zhang et al., 2024) typically fix images to a 4:1 aspect ratio, which may distort the text structure of long or vertical text. We divide training samples into buckets based on text length and orientation. Samples within each bucket have similar length and layout characteristics. During training, each batch samples only from a single bucket, effectively mitigating performance degradation in complex layouts.

### 3.4 BENCHMARK DATASET

**CTR-X.** We extend the existing scene text benchmark CTR-TSR-Test (Chen et al., 2021b) to construct CTR-X, aiming to systematically evaluate text super-resolution across diverse text lengths and orientations. Specifically, we group images in the original CTR-TSR-Test dataset by aspect ratio and randomly sample up to 50 per group, taking all if fewer, to ensure sufficient vertical text samples. In total, CTR-X contains around 1500 samples, covering aspect ratios from 1:15 to 15:1. All images and annotations are manually checked and corrected to ensure high image quality and label accuracy. Thus, CTR-X serves as a standardized benchmark for assessing model robustness in variable-length text, vertical layouts.

**RareText.** There is currently a lack of real-world Traditional Chinese text images for evaluation. Existing datasets mainly target classical texts (Saini et al., 2019), with limited scene diversity or low resolution (Chen et al., 2021c), making them unsuitable for text image super-resolution tasks. To

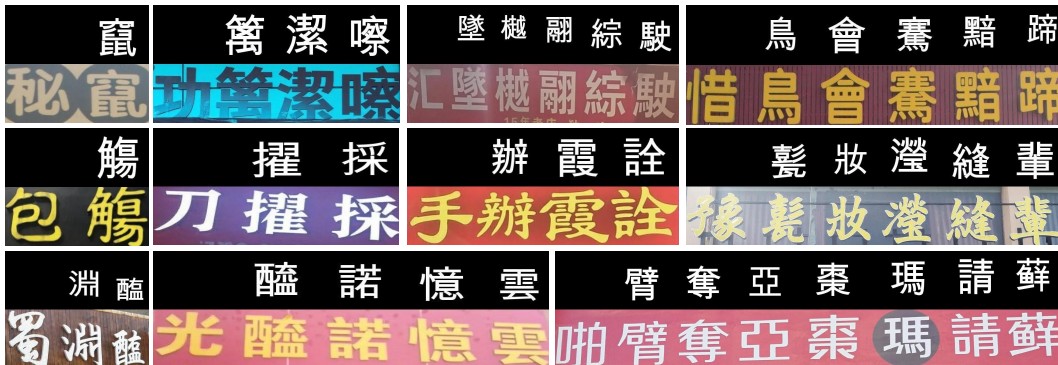

Figure 3: Text image generation results of our proposed Gen-Text.

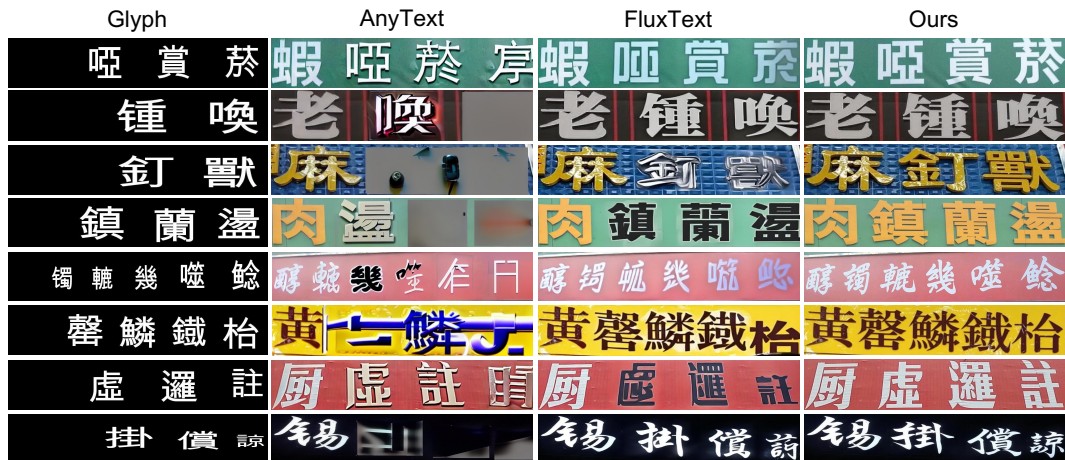

Figure 4: Text generation results compared to existing methods.

address this, we construct RareText, providing a challenging benchmark for assessing models' ability to reconstruct Traditional and complex Chinese characters in real-world scenarios. The dataset comprises about 1,000 high-resolution images collected from diverse environments, covering a wide range of character types, scene conditions, and font styles. It contains: (i) Traditional Chinese characters, characterized by dense radicals and strokes, commonly used in Taiwan, Hong Kong, Macau, and classical literature; (ii) complex characters with more than 15 strokes, intricate structures, or artistic glyphs derived from inscriptions, ancient scripts, and stylized fonts. This design enables comprehensive evaluation of stroke-intensive, artistic, and non-standard layouts. Images are sourced from social media, forums, cultural archives, and open repositories. Annotations are first generated using PPOCR-v5 (Cui et al., 2025) and then manually verified to ensure high-quality character-level labels and accurate text orientation.

## 4 EXPERIMENTS

### 4.1 IMPLEMENTATION DETAILS

**Text Image Generation.** We first construct training data on the CTR dataset. Specifically, we use OCR to detect character positions in each image and extract the semantic information of each character, generating position images and glyph images as input features. For each image, the first character is retained as a style reference to encode font, stroke thickness, and structural style information. During training, we perform LoRA fine-tuning of Gen-Text on the FLUX-Fill (Labs, 2024) dataset to improve the model's adaptability and generation quality on real-world text images.

**Text Image Restoration.** We strictly follow the DiffTSR (Zhang et al., 2024) pipeline to preprocess the CTR dataset: *i) remove the images with a resolution smaller than 64 pixels; ii) only retain images with a width-to-height ratio greater than 2; iii) only retain images with the length of text annotations*

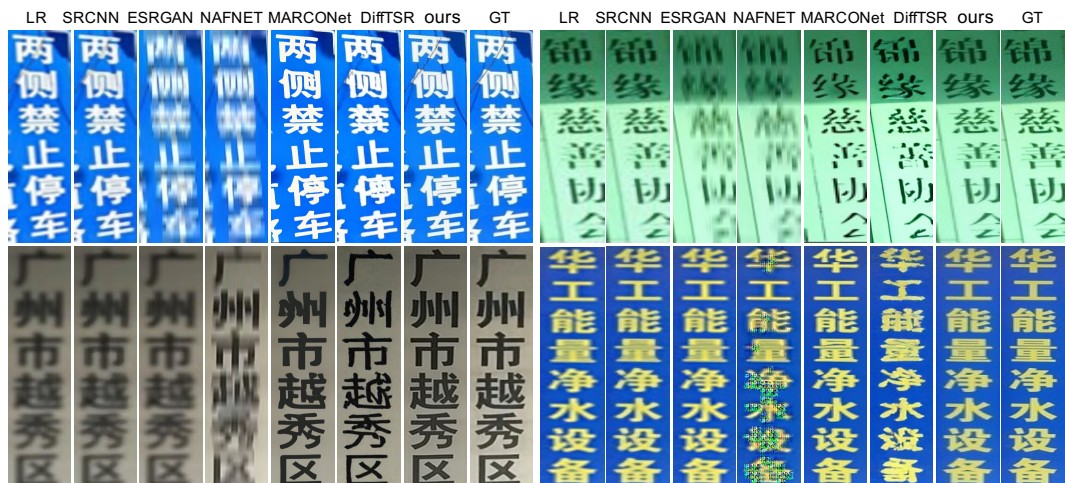

Figure 5: Qualitative results of different methods on vertical text.

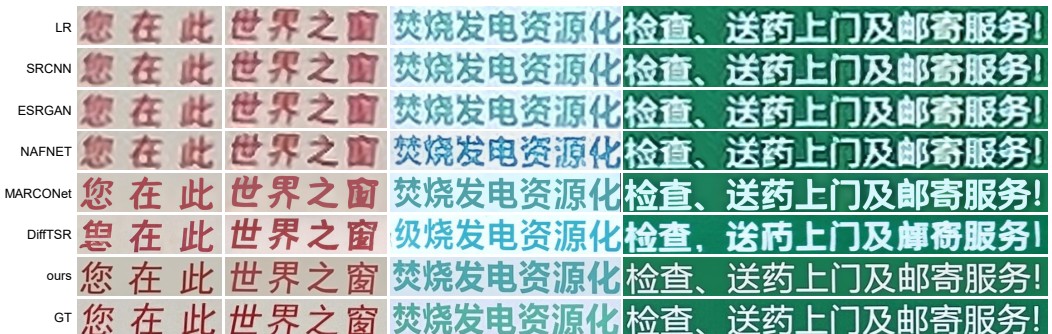

Figure 6: Qualitative comparison for the real-world dataset RealCE with different methods.

*not larger than 24.* In addition, we augment the training set with high-quality Chinese text images generated by Gen-Text. Low-resolution (LR) images are synthesized via Real-ESRGAN (Wang et al., 2021) degradation to simulate real-world image distortions. During training, the diffusion network adopts a U-Net architecture, with the upsampling module combining SwinIR (Liang et al., 2021) to simultaneously improve text structure and visual clarity.

More detailed information can be found in the Appendix.

## 4.2 QUANTITATIVE COMPARISON

We conduct a systematic evaluation of existing competitive methods, including SRCNN (Dong et al., 2015), ESRGAN (Wang et al., 2018), NAFNET (Chen et al., 2022), TSRN (Wang et al., 2020), TBSRN (Chen et al., 2021a), TATT (Ma et al., 2022), MARCONet (Li et al., 2023) and DiffTSR (Zhang et al., 2024). The evaluation covers four benchmarks: CTR-TSR-Test, Real-CE, CTR-X, and RareText. For TSR-TSR-Test (Chen et al., 2021b) and Real-CE (Ma et al., 2023c), we directly adopt the results reported in DiffTSR (Zhang et al., 2024), while for CTR-X and RareText, we strictly follow the official codes and released weights to ensure fair and comparable results.

As shown in Tab. 1 and Tab. 2, Re-Text consistently outperforms existing competitors across all benchmarks in terms of both image quality and text accuracy. Specifically, our method achieves improvements of 4.88%, 2.12%, 6.88%, and 4.9% in PSNR over the latest approaches (Zhang et al., 2024), and gains of 7.08%, 3.21%, 5.19%, and 14.05% in OCR accuracy. These results validate the superiority of our approach and demonstrate its robustness and generalization across diverse and challenging scenarios.

Table 3: Impact of data generated by Gen-Text on the effectiveness of super-resolution methods.

| Methods | PSNR↑ | LPIPS↓ | FID↓ | ACC ↑ |
|---|---|---|---|---|
| DiffTSR w/o *Gen-Text* | 17.49 | 0.336 | 70.59 | 0.8475 |
| DiffTSR w/ *Gen-Text* | 19.09 | 0.355 | 98.17 | 0.8696 |
| Our w/o *Gen-Text* | 19.56 | 0.322 | 59.30 | 0.8772 |
| Our w/ *Gen-Text* | 19.61 | 0.3011 | 55.49 | 0.8794 |

Table 4: Effectiveness verification of different modules in Re-Text.

| Diffusion | GAN | CTR-X | | | | | RealCE | | | | |
|---|---|---|---|---|---|---|---|---|---|---|---|
| | | PSNR↑ | SSIM↑ | LPIPS↓ | FID↓ | ACC↑ | PSNR↑ | SSIM↑ | LPIPS↓ | FID↓ | ACC↑ |
| *Only* | *None* | | | | | *Out of Memory* | | | | | |
| *None* | *Only* | 23.03 | 0.677 | 0.253 | 52.01 | 0.5836 | 18.28 | 0.58 | 0.298 | 95.84 | 0.8490 |
| *Down×4* | *None* | 25.42 | 0.745 | 0.295 | 55.21 | 0.6276 | 19.63 | 0.63 | 0.412 | 97.21 | 0.8732 |
| *Down×4* | *Up×4* | 26.97 | 0.788 | 0.165 | 29.05 | 0.6468 | 19.61 | 0.63 | 0.301 | 55.49 | 0.8791 |

## 4.3 QUALITATIVE COMPARISON

**Text Image Generation.** We conduct a qualitative analysis to demonstrate the capability of Gen-Text in generating complex text images, as shown in Fig. 3 and Fig. 4. Fig. 3 presents diverse samples generated by Gen-Text, including various fonts and text lengths. The results show that Gen-Text preserves fine-grained stroke structures while producing visually realistic and stylistically diverse text images. Fig. 4 compares Gen-Text with existing AnyText (Tuo et al., 2023) and FluxText (Lan et al., 2025). Compared to these methods, Gen-Text more accurately recovers complex glyphs, such as stroke-dense Traditional Chinese characters and artistic fonts, maintaining stroke integrity and font details. Backgrounds are also more naturally rendered, closely resembling real-world scenes. Overall, the visual analysis demonstrates that Gen-Text effectively generates structurally accurate, style-diverse, and visually realistic text images, providing high-quality and rich training samples for downstream text super-resolution models.

**Text Image Restoration.** We further conduct a visual comparison of Re-Text with representative text super-resolution methods, as shown in Fig. 1, 5. It can be observed that Re-Text produces more precise strokes, particularly for long texts, vertical layouts, and complex characters. In contrast, traditional methods (Dong et al., 2015; Wang et al., 2018; Chen et al., 2022) struggle with severely degraded inputs, resulting in poor overall restoration. MARCONet (Li et al., 2023) can recover text structure but often suffers from inconsistent styles and text-background separation. DiffTSR (Zhang et al., 2024), while partially effective in restoring low-resolution text, exhibits inferior background restoration and struggles with long or vertically arranged text. By combining a bucket training strategy with the strengths of both diffusion and GAN, Re-Text generates high-resolution images with superior text fidelity and consistent visual quality. Moreover, Fig. 6 illustrates the fine-tuned results of Re-Text on the RealCE dataset, demonstrating its strong adaptability and robustness under real-world degradations.

See the Appendix for more visualizations.

## 4.4 ABLATION STUDY

**Effectiveness of the Gen-Text.** We first evaluate the contribution of each module in Gen-Text through a systematic ablation study on the glyph feature, style feature, and Sobel loss, as shown in Fig. 7. **Glyph only:** The generated text preserves the basic structure of complex characters, but fine-grained control over strokes is limited, and font styles appear random. **Glyph+Style:** Incorporating the style features produces text with more consistent font styles, and stroke thickness and curvature are better controlled. **Glyph+Style+Sobel loss:** Adding the Sobel loss further enhances stroke sharpness and edge details, achieving the best performance in terms of structure, style, and fine details. Additionally, we also evaluate the impact of high-quality data generated by Gen-Text on text super-resolution tasks (Tab. 3). The results show that these high-quality synthetic images significantly improve text recognition accuracy. They also enhance visual quality metrics such as PSNR and FID, partially compensating for the low quality of the original datasets.

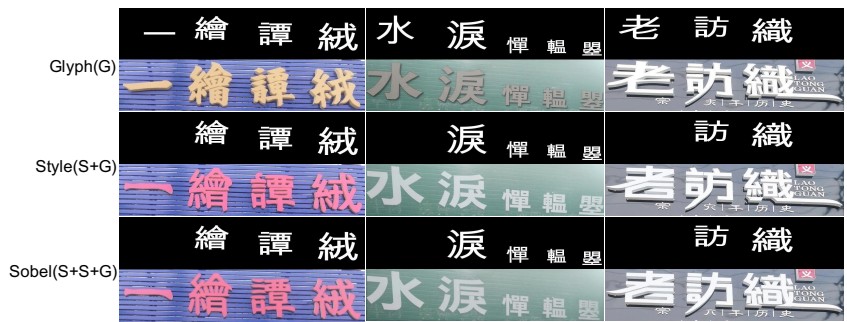

Figure 7: Visual analysis of different modules in Gen-Text.

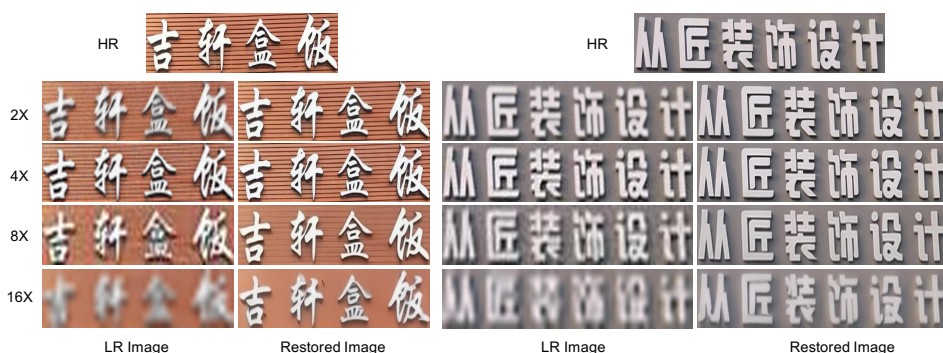

Figure 8: Qualitative analysis under higher degradation scales.

**Effectiveness of the Re-Text.** To validate the effectiveness of Re-Text, we design three variants for ablation study: *GAN-only:* training a GAN without diffusion. *Downsampling+Diffusion:* downsampling the input by 4×, applying diffusion restoration, and then upsampling back to the original size. *Downsampling+Diffusion+GAN (Re-Text)*: downsampling the input by 4×, restoring global structures with diffusion, and further enhancing stroke fidelity and texture details with GAN. The experimental results show that: Only the GAN module struggles to generate complex characters due to its limited generative capacity. Due to GPU memory limitations, Diffusion-only cannot directly handle 512x128(or larger) LR inputs, making it impractical in real-world scenarios. Diffusion with downsampling alleviates the memory bottleneck and improves recognition accuracy; however, its perceptual quality (LPIPS) is even worse than that of only-GAN. Re-Text combines diffusion for structural restoration with GANs for fine-grained refinement, preserving structure while enhancing visual details, thereby achieving superior visual quality.

**Effectiveness under more severe degradations.** We also conduct a qualitative visual analysis of the model's performance under more severe degradations. Specifically, the images are progressively downsampled by factors of 2, 4, 8, and 16. As shown in Fig. 8, even under the extreme 16× degradation, our model is still able to restore the images, with text structures clearly preserved. The results highlight the model's robustness under severe degradation and its potential for real-world super-resolution.

## 5 CONCLUSION

In this work, we present ReGenText, a systematic framework for diverse text image super-resolution. Unlike previous methods that focus solely on model design, ReGenText integrates data generation, image restoration, and training strategies to address data scarcity, limited model fidelity, and complex layout adaptation. Specifically, Gen-Text jointly models font styles and glyph representations to synthesize large-scale, high-quality text images. Re-Text combines the generative capabilities of diffusion models with the detail-enhancing strengths of GANs, improving both structural fidelity and visual clarity. The bucket training strategy further enhances the robustness to variable-length and vertical texts. Extensive experiments on the CTR-TSR-Test, Real-CE, CTR-X, and RareText benchmarks demonstrate significant improvements in OCR accuracy and visual quality. These results validate the effectiveness of ReGenText in real-world scenarios.

## REPRODUCIBILITY STATEMENT

We provide implementation details in Sec. 4.1 and Appendix A.2, including the training process and selection of hyper-parameters. We also provide details on dataset preparation in Sec. 3.4, and the code and data will be made available along with it.

## ETHICS STATEMENT

This work focuses on text image super-resolution (TISR) and does not involve the collection of personally identifiable or sensitive data. All experiments use publicly available benchmarks and synthetically generated datasets. While our method can improve accessibility in applications such as digitization of historical texts and assistive technologies, it may also be misused for purposes such as document forgery or disinformation. We emphasize that our contributions are intended solely to advance research in text clarity and accessibility. Finally, we acknowledge the environmental cost of training large generative models and encourage energy-efficient practices and dataset reuse to reduce computational overhead.

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

## A  APPENDIX

### A.1  LLM ACKNOWLEDGEMENT

We thank ChatGPT for its assistance in revising our paper. All wording and factual content were reviewed, checked and approved by the authors.

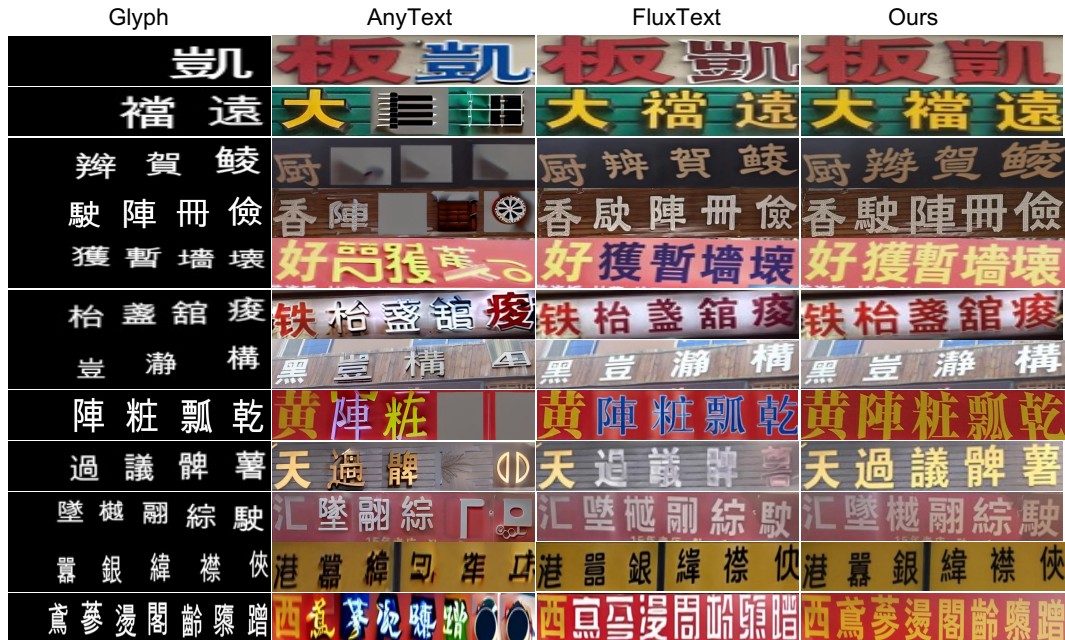

Figure 9: Qualitative results compared with other text image generation methods.

## A.2 MORE IMPLEMENTATION DETAILS

We trained two models for different text image tasks. For text generation, a LoRA-based model was fine-tuned on cropped text-line images with masked inputs, building on FluxFill base model. Training employed bucket-based sampling to accommodate varying aspect ratios, with a per-GPU batch size of 1 and gradient accumulation of 8, using the AdamW optimizer at a learning rate of $2 \times 10^{-5}$ and LoRA rank 256. Training ran for up to 3,0000 steps. For text image super-resolution, the model with a SwinIR (Liang et al., 2021) decoder was trained on high-resolution text images using 4 NVIDIA GPUs in a distributed setting via the HuggingFace accelerate framework. Training lasted 10 epochs with a per-GPU batch size of 8 (effective batch size 32), using the Adam optimizer with an initial learning rate of $1 \times 10^{-4}$. To improve robustness to extremely elongated text, training images were sampled with aspect ratios of 15:1 and 1:15. In the super-resolution task, we set the diffusion time steps $t$ to 1000 during training, and 20 during inference. This unified setup ensures reproducible and efficient training across both low-resolution text generation and high-resolution restoration tasks.

## A.3 EXAMPLES OF OUR SYNTHETIC DATASET

In Fig. 9, 10, 11, we present additional examples generated by Gen-Text and compare them with recent state-of-the-art methods. It can be observed that Gen-Text consistently produces text images with precise structures and diverse styles. This provides a continuous source of high-quality training data for subsequent super-resolution tasks, thereby effectively enhancing model robustness and performance.

## A.4 MORE COMPARISON RESULTS ON TEXT IMAGE SUPER-RESOLUTION

We provide additional qualitative results in Fig. 12 ,Fig. 13 and Fig. 14. In Fig. 12, we present further comparison results on the performance of complex characters. Our method faithfully reconstructs the text structures, benefiting from the use of synthetic data and the bucketed training strategy. In Fig. 13 and Fig. 14 , we show qualitative results for text images with various aspect ratios. Our method demonstrates consistent performance across different aspect ratios, highlighting its robustness.

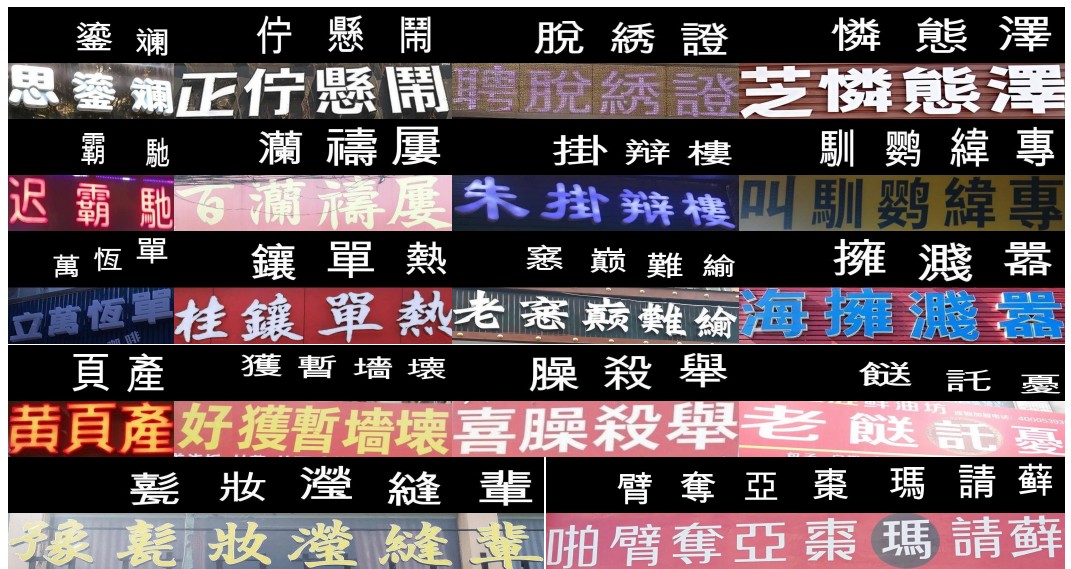

Figure 10: Examples of synthetic data generated by our text generation method.

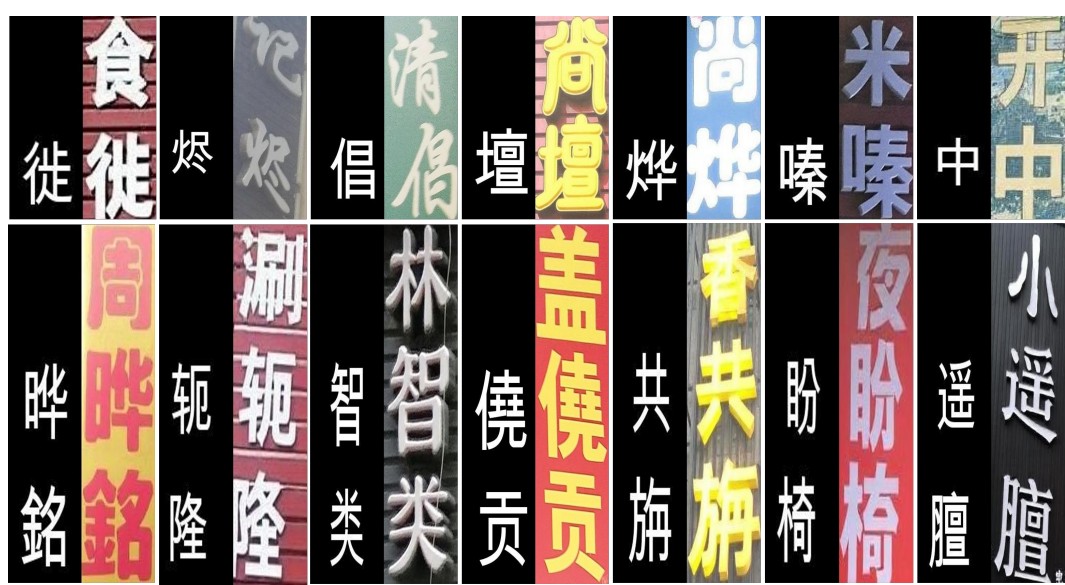

Figure 11: Qualitative results of vertical text image generation.

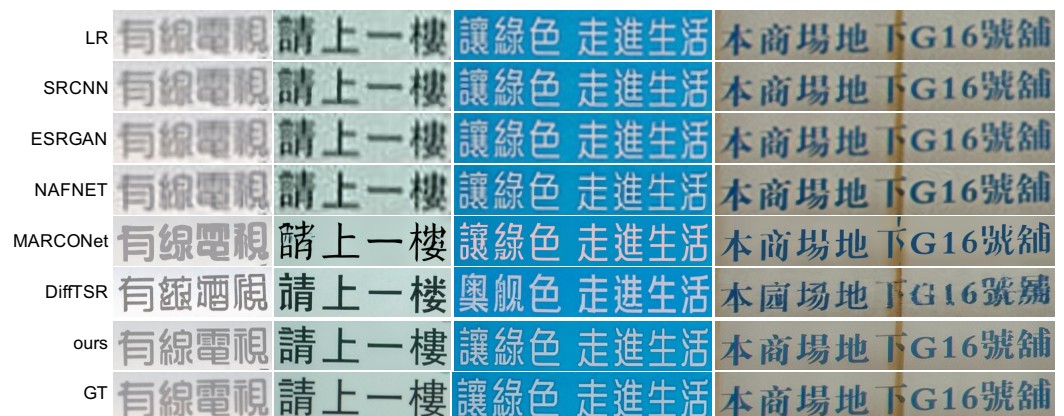

Figure 12: Qualitative results on complex characters.

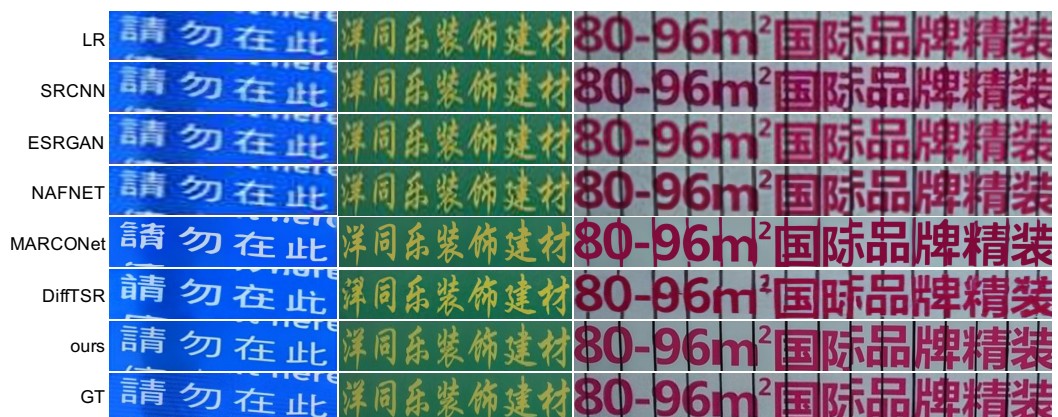

Figure 13: Qualitative results on various lengths.

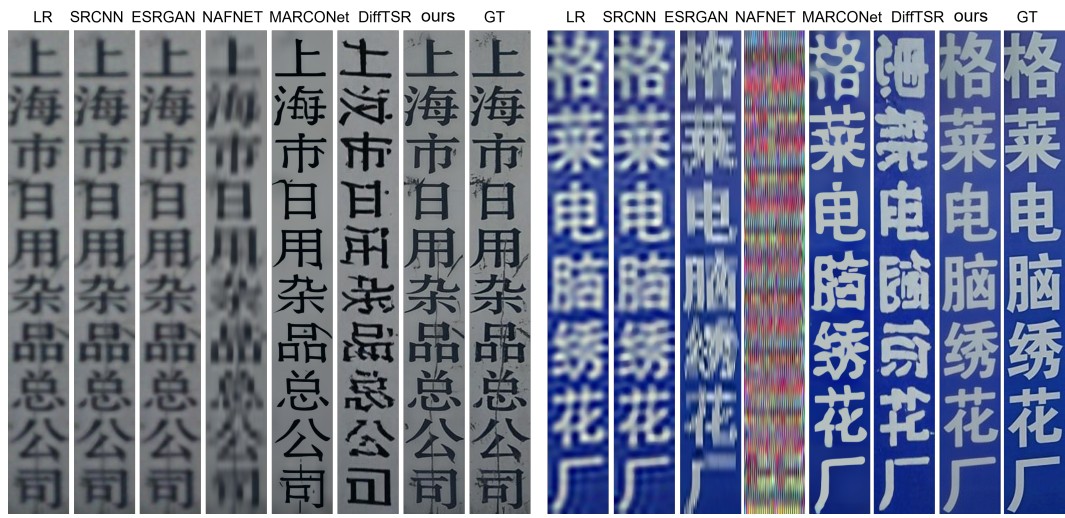

Figure 14: Qualitative results on vertical text.

