# OpenReview forum: "ReGenText: Joint Generation and Restoration for Diverse Text Image Super-Resolution"
_ICLR.cc/2026/Conference — ICLR 2026 Conference Withdrawn Submission_

### Official Review · Reviewer_455A · 2025-10-21

**Soundness:** 2
**Presentation:** 3
**Contribution:** 2
**Rating:** 4
**Confidence:** 3

**Summary:**

This paper introduces ReGenText, a comprehensive framework for text image super-resolution targeting visually degraded or diverse layouts in scene text images. The proposed solution is multi-pronged: (1) Gen-Text, a diffusion-based data generator coupling font style with glyph priors to address data scarcity and diversity; (2) Re-Text, a hybrid diffusion-GAN architecture aimed at restoring both structure and fine details in degraded text images; and (3) a bucket training approach that groups images by text length and orientation to improve handling of variable-length and vertical text. Experiments span standard and newly constructed datasets, showing improvements over prior work in both recognition accuracy and perceptual quality.

**Strengths:**

1. The authors do a credible job of addressing multiple bottlenecks in TISR—namely, data scarcity, model fidelity, and layout complexity—by presenting the Gen-Text, Re-Text, and Bucket Training modules within a unified system. This system-level perspective is a notable step up from works focusing solely on model architecture.

2.  Experiments are performed across a variety of challenging benchmarks, including standard (CTR-TSR-Test, Real-CE) and new (CTR-X, RareText) datasets. Table 1 and Table 2 back up claims of performance, and ablation in Table 3 provides insight into Gen-Text's contribution. Across the figures (notably Figure 1, Figure 5, Figure 12–14), the qualitative superiority and generalization to diverse layouts, including vertical and rare characters, are clearly demonstrated.

**Weaknesses:**

1. The authors identify three key challenges: limited training data, model fidelity, and adaptability to complex layouts. However, these issues are long-standing and have been extensively explored in recent literature, with numerous methods proposed to mitigate each of them, e.g. TATT to the complex layouts. The manuscript provides limited discussion or comparison with these directly related works in the Introduction Section.
2.  While the bucket training idea (Section 3.3) is promising, there is a lack of concrete experimental analysis quantifying its direct impact versus baselines, either through dedicated ablation or detailed reporting. Which of the observed improvements (e.g., in Table 2 on vertical/long text layouts) are attributable to bucket training rather than other system modules? How are buckets chosen (granularity, limits) and do they trade off training time or convergence?
3. Nowhere is there substantive analysis of the system’s failure cases, especially for rare, artistic, or heavily occluded scripts. The qualitative figures do show strong results but fail to include explicit examples of breakdown or unexpected behavior.
4. The authors claim strong generalization to rare characters and variable layouts, but their model’s training, as stated in the Appendix (Page 13–14), relies heavily on synthesized data covering these variations. More detail is needed on how the model performs on genuinely out-of-distribution cases or unseen scripts beyond the constructed RareText.
5.  While the evaluation is thorough, some direct competitors for synthetic data generation (e.g., UDiffText, Glyph-ByT5, GlyphDraw) are not quantitatively compared in the text-image synthesis setting. This limits claims about Gen-Text’s relative effectiveness in training data generation compared to other advanced methods.

**Questions:**

1. How would the method perform if Gen-Text did not have access to similar fonts or glyphs as test data? In other words, is there a risk of overfitting synthetic data priors to distributions too similar to curated test sets?
2. Are there examples where ReGenText fails to restore text accurately, especially under combinations of rare characters, unusual artistic fonts, and severe blur? If so, can the failures be characterized (e.g., by text type, font, or degradation severity)?

---

### Official Review · Reviewer_aYHn · 2025-10-26

**Soundness:** 3
**Presentation:** 3
**Contribution:** 3
**Rating:** 6
**Confidence:** 4

**Summary:**

The paper presents ReGenText, a holistic pipeline for diverse text-image super-resolution (TISR). It combines three components:
1. Gen-Text – a diffusion Transformer that synthesises high-fidelity text images by conditioning on font style and glyph priors;
2. Re-Text – a hybrid diffusion–GAN architecture that first denoises down-sampled latents and then refines them adversarially for sharp strokes;
3. Bucket Training – grouping mini-batches by text length/orientation to avoid aspect-ratio distortion.
Two new benchmarks (CTR-X for long/vertical text, RareText for complex Traditional-Chinese characters) are introduced.
Extensive experiments show >4 dB PSNR gains and +7% OCR-accuracy improvements over prior arts.

**Strengths:**

1. Gen-Text generated data consistently boosts SR metrics (Table 3) and can benefit any downstream TISR method.
2. Thorough evaluation: four datasets, five metrics, statistical significance reported; visual results convincingly show clearer strokes and less hallucination.

**Weaknesses:**

1. Methodological novelty is incremental. Gen-Text extends AnyText/Flux-Text with Sobel loss; Re-Text is essentially DiffTSR + GAN post-refinement; Bucket Training is standard aspect-ratio bucketing.
The paper does not fundamentally advance diffusion theory or architectural design.
2. Missing baselines with other powerful SR backbones. Comparison is limited to SRCNN/ESRGAN/NAFNet/DiffTSR; no SwinIR, and Real-ESRGAN are re-implemented under the same training data for fair comparison.
3. The experimental comparison stops at DiffTSR (CVPR 2024) and omits several latest diffusion-based SR works that have shown superior performance or efficiency, e.g.: DiT4SR, FaithDIff, PiSR-SR, TSD-SR, and TSD-SR. The authors should provide comparative results against these methods.

**Questions:**

1.Why do LPIPS and FID become worse after DiffTSR is augmented with Gen-Text in Table 3?
2. Why does DiffTSR exhibit a much larger PSNR improvement than ours in Table 3?

---

### Official Review · Reviewer_UgZp · 2025-11-01

**Soundness:** 3
**Presentation:** 3
**Contribution:** 2
**Rating:** 6
**Confidence:** 4

**Summary:**

The paper introduces ReGenText, a unified framework that addresses the key challenges in Text Image Super-Resolution (TISR) — namely limited training data, low model fidelity, and poor adaptability to complex text layouts. Unlike prior approaches that focus solely on model architecture, ReGenText integrates data generation, image restoration, and layout-aware training into a single, systematic solution. Additionally, the authors construct two new benchmarks: CTR-X (for variable-length and vertical text) and RareText (for complex and traditional Chinese characters), providing comprehensive evaluations of robustness and generalization. Extensive experiments on standard and newly proposed benchmarks demonstrate that ReGenText achieves state-of-the-art performance, surpassing existing methods like DiffTSR and MARCONet in PSNR, FID, LPIPS, and OCR accuracy.

**Strengths:**

1. The Gen-Text module introduces a diffusion-based generator guided by glyph priors and font style embeddings, producing high-quality and diverse synthetic text images. This component effectively alleviates data scarcity and improves model robustness, particularly for rare and complex characters, which are underrepresented in existing datasets.

2. The Re-Text model innovatively combines diffusion models (for structural consistency) and GANs (for realistic textures and sharpness), achieving a balanced trade-off between structural precision and perceptual quality. Also, the Bucket Training strategy—grouping samples by text length and orientation—tackles a long-standing limitation of TISR models, which often fail on long or vertically oriented text.

3. The authors go beyond existing datasets by introducing two new benchmarks, namely CTR-X and RareText, to fill critical evaluation gaps in TISR and will likely serve as valuable resources for future research.

**Weaknesses:**

1. The proposed framework integrates diffusion models, GANs, and transformer-based components, which may together make ReGenText computationally expensive to train and deploy.

2. For the data generation part, the ablation study shows that the generated samples from Gen-Text improve the performance of TISR task, but lacks the quantitative analysis of this module on the quality of the generated data.

3. For the image super-resolution part, this paper leverages diffusion model to restore low-resolution images, and uses GAN to control the prediction of diffusion model. It remains not clear whether GAN helps “improve sharpness and perceptual realism” or just provide supervision.

4. There remain some typos, like “allevi- ating” in line 88.

**Questions:**

1. The Gen-Text module for data augmentation seems not too complex, and achieves good results for TISR as shown in Table 3. However, this paper didn’t analyze the effectiveness of each design in this part. Although Figure 7 shows some visual results, it can be beneficial to make quantitative analysis on this module to show the rationality of module design.

2. This paper uses diffusion model together with GAN to control the generation of super-resolution images. It is necessary to show the effectiveness of GAN in the improvement of sharpness and perceptual realism. The authors can also compare Re-Text with other methods like ControlNet to show the advantages of their solution.

3. Text prediction from OCR is engaged by cross attention to lead the training of Gen-Text. How will the quality of OCR model impact the performance of Gen-Text?

4. It is also advised to evaluate the computational complexity of this method during training and inference stage.

---

### Official Review · Reviewer_DH2P · 2025-11-01

**Soundness:** 2
**Presentation:** 3
**Contribution:** 2
**Rating:** 2
**Confidence:** 5

**Summary:**

The paper proposes ReGenText, a systematic framework for Text Image Super-Resolution (TISR) that addresses three key challenges: limited training data, insufficient visual fidelity, and poor adaptability to complex text layouts (e.g., long or vertical text). ReGenText integrates three core components:
- Gen-Text: a diffusion-based data generation module that synthesizes high-quality, diverse text images using font style and glyph priors to alleviate data scarcity.
- Re-Text: a hybrid diffusion–GAN restoration model that combines the structural accuracy of diffusion models with the fine-detail enhancement of GANs for high-fidelity reconstruction.
- Bucket Training: a layout-aware training strategy that groups samples by text length and orientation to improve generalization on variable-layout texts.

The authors also introduce two new benchmarks—CTR-X (for variable-length and vertical text) and RareText (for rare/complex Chinese characters).

**Strengths:**

- The proposed Re-Text model skillfully combines diffusion and GAN paradigms, achieving superior balance between structural accuracy and visual realism.
- The introduction of CTR-X and RareText datasets enables more rigorous and realistic evaluation of TISR methods on challenging, real-world scenarios like vertical text and rare characters.
- Strong technical execution and clear writing.

**Weaknesses:**

- The core idea of coupling data generation and restoration is not entirely new; for instance, [1] already proposed a unified framework using three diffusion models for generate–degrade–restore training, which diminishes the conceptual novelty of the proposed pipeline.
- The low-resolution images are synthesized, which may not fully capture the diversity and complexity of real-world degradations (e.g., motion blur, compression artifacts, non-uniform noise), potentially limiting practical applicability.
- While modules are evaluated, the individual contribution of bucket training to final performance (e.g., on vertical vs. long text) is not quantified separately, making it hard to assess its standalone impact.
- Re-Text uses OCR-derived text priors for guidance; errors in OCR (especially on severely degraded inputs) could negatively affect restoration quality, yet robustness to OCR noise is not analyzed.
- The method is not evaluated on TextZoom, a widely adopted benchmark for text image super-resolution, raising concerns about comparability with existing TISR literature.

[1] Scene Text Image Super-resolution based on Text-conditional Diffusion Models, WACV 2024

**Questions:**

Please refer to Weaknesses.

---

### Note · Authors · 2025-11-14

I have read and agree with the venue's withdrawal policy on behalf of myself and my co-authors.